# Assessing the Test-Retest Reliability of MyotonPRO for Measuring Achilles Tendon Stiffness

**DOI:** 10.3390/jfmk10010083

**Published:** 2025-02-28

**Authors:** Krystof Volesky, Jan Novak, Michael Janek, Jakub Katolicky, James J. Tufano, Michal Steffl, Javier Courel-Ibáñez, Tomas Vetrovsky

**Affiliations:** 1Faculty of Physical Education and Sport, Charles University, 162 52 Prague, Czech Republic; k.volesk@gmail.com (K.V.);; 2Department of Rehabilitation and Sports Medicine, Second Faculty of Medicine, University Hospital in Motol, Charles University, 150 06 Prague, Czech Republic; 3Department of Physical Education and Sport, Faculty of Sport Sciences, University of Granada, 520 71 Melilla, Spain

**Keywords:** Achilles tendon, stiffness, reliability, MyotonPRO, tendon

## Abstract

**Objectives**: This study evaluates the test-retest reliability and inter-rater reliability of the MyotonPRO for measuring Achilles tendon stiffness at two standardized sites over various time frames and settings. **Methods**: Eight healthy participants underwent assessments by three raters over six visits. Tendon stiffness was measured at proximal (mid-portion) and distal (insertional) regions of the Achilles tendon at various time frames (10–15 s, 10–15 min, 24 h, and 14 days apart). Measurements included participant repositioning and two activity stimuli (daily living and sport). Reliability was calculated using the intraclass correlation coefficient (ICC), its 95% confidence interval, coefficient of variation, standard error of measurement, and minimal detectable change. **Results**: Short-term reliability (10–15 min) was excellent, with an ICC of 0.956 (0.929–0.974). Between days reliability (24 h) was good, with an ICC of 0.889 (0.802–0.938). Between weeks reliability (2 weeks) was good with an ICC of 0.886 (0.811–0.931). Short-term reliability with the simulation of activity of daily living was good, with an ICC of 0.917 (0.875–0.945). Short-term reliability with the simulation of sport was good with an ICC of 0.933 (0.891–0.96). Between days reliability with the simulation of sport was good, with an ICC of 0.920 (0.859–0.955). **Conclusions**: When used in a standardized position, the MyotonPRO demonstrates reliable repeated measurements of Achilles tendon stiffness. This protocol provides a foundation for clinical research and rehabilitation by clarifying expected reliability across minutes, days, and weeks, thus aiding clinicians and researchers in monitoring tendon adaptations and making evidence-based decisions.

## 1. Introduction

The Achilles tendon transmits force and absorbs energy during activities such as walking, running, and jumping [1] and is subject to extremely high mechanical loads, often up to ten times the body weight [2]. In athletes, excessive mechanical loading can lead to Achilles tendinopathy, a condition characterized by tendon pain and loss of function [3]. Its prevalence among athletes stands at 6%, affecting both men and women equally [4]. Although Achilles tendinopathy can be an acute injury, it often evolves into a chronic condition that can impair quality of life and work productivity [5]. A significant number of affected individuals experience persistent symptoms for years, resulting in reduced physical activity levels [6].

Considering the prevalence and consequences of Achilles tendinopathy, understanding the biomechanical properties of the Achilles tendon, namely its elasticity and stiffness, is essential for assessing its functional dynamics and vulnerability to injury. Elasticity enables the tendon to store energy in a spring-like manner, while stiffness reduces the extent of elongation, protecting collagen fibers against damage. In general, tendons can elongate by up to 4% of their length without sustaining any damage. However, if the resistance to elongation is inadequate, elongation between 4 and 8% may lead to the breakdown of collagen cross-links [7]. This can lead to structural changes that contribute to the development of tendinopathy. When elongation exceeds a critical threshold (above 8%), collagen fibers may undergo macroscopic failure, potentially leading to Achilles tendon rupture and complete loss of function. Thus, the stiffness of the tendon, in conjunction with the strength of the triceps surae muscle, is critical in determining the level of resistance to elongation and in preventing excessive elongation that could damage collagen fibers [8].

There are several methods for measuring Achilles tendon stiffness. For example, research involving healthy subjects has often relied on calculations that quantify tendon displacement (Δmm) during maximal voluntary contraction of plantar flexor muscles [9]. However, this approach may not be suitable for tendinopathy cases, as central inhibition of plantar flexors can limit true maximal voluntary contraction. Thus, to assess stiffness in individuals who present tendinopathy, shear wave ultrasound elastography (SWUE) has been commonly employed [10]. Despite studies showing good reliability of SWUE, its reliability is highly dependent on the operator and requires extensive ultrasound expertise to accurately identify the structures and artifacts being assessed [11]. Consequently, there is a need for more accessible techniques that can reliably measure tendon stiffness in tendinopathy patients in both research and clinical settings [12].

The MyotonPRO is a promising tool for quantifying the stiffness of the Achilles tendon in patients with tendinopathy, where a force is applied transversely to the tendon fiber axis, and the resultant displacement of the tendon tissue is measured. Originally designed for assessing skeletal muscles, the MyotonPRO is a portable device that employs a controlled preload of 0.18 N to compress the subcutaneous tissue, followed by a 15 ms impulse of 0.40 N of mechanical force, which elicits a damped or decaying natural oscillation within the tissue, enabling the measurement of tendon stiffness [13]. Compared to more operator-dependent modalities (e.g., ultrasound elastography), the MyotonPRO requires less specialized training and reduces user-dependent variability. Its portability and straightforward setup make it a convenient tool for researchers and clinicians to measure soft-tissue stiffness quickly and accurately. To determine a device’s reliability, it is critical to consider the magnitude of measurement errors in absolute values after repeated conditions. In practical terms, absolute errors are essential to identify whether the changes in tendon stiffness observed after a given intervention are due to the actual changes in the functional dynamics of the athletes (adaptations) [14].

Several key factors influence the reliability of measuring tendon stiffness via the MyotonPRO. First, the accuracy of the results is highly contingent on the precise location of the measurement and the participant’s positioning [15]. Therefore, standardizing both the device’s position and the participant’s posture is essential for ensuring consistent, repeatable measurements in research and clinical practice. However, existing studies evaluating the MyotonPRO’s reliability lack straightforward recommendations for standardized positioning. Second, temporal fluctuations in Achilles stiffness are particularly relevant in clinical settings and in monitoring patients with tendinopathy [16]. Moreover, physical activity stimuli like walking, biking, or sports practice often precede or occur between measurements, making the understanding of their impact vital for interpreting clinical stiffness measurements. Yet, studies reporting on MyotonPRO’s reliability have neither standardized device and patient positioning [17,18,19] nor accounted for various time intervals and stimuli between measurements [20,21].

Considering the lack of established reliability data, the objective of this study was to (1) assess the test-retest reliability of the MyotonPRO in measuring Achilles tendon stiffness using a newly established standardized position, (2) determine whether different time intervals (ranging from seconds to weeks) influence measurement reliability, and (3) evaluate how physical activity stimuli (activities of daily living and sport-like exercises) affect short-term reliability. Furthermore, (4) we explored the inter-rater reliability of the MyotonPRO in this context.

We hypothesized that, under standardized conditions, the MyotonPRO would demonstrate good to excellent test-retest reliability (intraclass correlation coefficient > 0.75) across all assessed intervals and activity stimuli.

Regarding the time frames specifically, immediate measurements at 10–15 s apart allowed us to capture consecutive trials without any repositioning of the participant. The 10–15 min interval served as a short-term retest period, encompassing practical requirements such as repositioning participants, marking the measurement site, and, in some cases, performing a quick activity stimulus or rest. This slight variability (i.e., sometimes closer to 10 min, sometimes 15 min) reflects realistic conditions in clinical and research settings, where minor procedural or participant-related delays commonly occur.

By providing both a clear hypothesis and a rationale for the time intervals used, this study addresses a gap in the literature and offers a reproducible methodology for reliably assessing Achilles tendon stiffness in healthy and potentially clinical populations.

## 2. Materials and Methods

### 2.1. Sample

Eight healthy young adults were recruited for this study through convenience sampling at the Charles University campus between March and July 2022. The sample comprised six males and two females, with a median age of 27.5 years (IQR 3.5), median body mass of 75 kg (IQR 15.5), and median height of 178 cm (IQR 5.8). On average, they engaged in 5.5 h (IQR 3) of sports training per week.

To calculate the sample size required to estimate the ICC of 0.9 with the lower bound of 95% confidence interval greater than 0.5 (the threshold for moderate reliability), we used Zou’s formula, as implemented in ICC.Sample.Size package (version 1.0) in R. Assuming three ratings per participant, a desired power of 80%, and using a two-sided 0.05 significance level, the required sample size is 8 [22]. Because this was a focused reliability study, we did not perform a separate power analysis for each subset of reliability measures.

Eligibility criteria included being under the age of 30 years, engaging in regular physical activity, and having no history of lower limb injuries within the previous six months. Participants also had to be free from any neurological, vascular, or systemic diseases and possess a valid sports permit from a medical doctor.

These criteria yielded a relatively narrow age range (median: 27.5 years), which may limit generalizability to a broader population. Furthermore, although the MyotonPRO may be used in tendinopathy cases, our sample included only healthy participants.

This study was approved by the Ethics Committee of the Faculty of Physical Education and Sport of Charles University (ID: 255/2021), and written informed consent was provided by the participants.

### 2.2. Experimental Approach

This study comprised 3 waves of data collection, each spaced 14 days apart (Figure 1). Each wave consisted of 2 visits on 2 consecutive days. The 1st visit involved 3 sessions (session 1 to session 3), and the 2nd visit consisted of 2 sessions (session 4 and session 5). A set of standardized measurements (SM) was taken during each session, consisting of 3 measurements at two different points (proximal and distal) on each leg, resulting in 12 measurements per SM. Between session 1 and session 2, a protocol mimicking vigorous sports activities—simulation of sport activity (SP) was introduced, while a protocol—simulation of activities of daily living (ADL) was applied between session 4 and session 5. In summary, each participant underwent 180 measurements: 3 waves × 5 sessions × 2 legs × 2 points × 3 measurements each. Figure 1 illustrates the measurement timeline for an individual participant. These measurements were used to calculate test-retest reliability across various time frames (10–15 s, 10–15 min, 24 h, and 14 days apart), repositioning of the subject (standardized measurements consistency), and physical activity stimuli (ADL and SP; Table 1).

### 2.3. Rationale for Time Intervals

The 10–15 s interval enabled immediate consecutive measurements without repositioning, reflecting direct repeatability. The 10–15 min interval accounted for practical short-term repositioning and minor procedural tasks. The 24 h interval represented next-day follow-up, commonly employed in clinical check-ups. Finally, the 14-day interval allowed the assessment of stability over a longer period, relevant for monitoring rehabilitation programs.

### 2.4. Achilles Tendon Stiffness

The MyotonPRO (Muomeetria, Tallinn, Estonia; Model 000607) was used to collect data from the tendon stiffness (Appendix A). The device applies a controlled preload of 0.18 N to compress the subcutaneous tissue, followed by a 15 ms impulse of 0.40 N of mechanical force, which elicits a damped or decaying natural oscillation within the tissue, enabling the measurement of tendon stiffness [13].

### 2.5. Standardized Measurement

Standardized measurement started with participants sitting on a box with their buttocks fixed using a wedge (Figure 2A). The height of the box was set so that the angle at the ankle and knee was 90° as controlled by a goniometer, and the line connecting the medial side of the heel and big toe was adjusted to be perpendicular to the box. Achilles tendon stiffness (N/m) was measured at two specific points (Figure 2B): the distal point, situated 1 cm proximally to the tuber calcanei, and the proximal point, located 6 cm proximally to the tuber calcanei [23]. The MyotonPRO device was placed on the adjustable rack during measurement (Appendix A). These points were marked with a permanent marker lasting until the next day but not until the next wave of measurements. In this position, the distance between the big toe and the box, as well as the distance between the first metatarsi of the left and right feet, were recorded (Figure 2C). Finally, the MyotonPRO was underlaid by an adjustable rack, and the height of the rack for both distal and proximal measurement points was also recorded (Figure 2D).

### 2.6. Simulation of Sports and Activities of Daily Living

The simulation of the sports activities (SP) targeted a single leg using the HumacNorm dynamometer (Cybex 770 NORM ®, Humac, CA, USA). The targeted leg was chosen randomly for each participant, and this selection was maintained for all three waves. The simulation involved loading the Achilles tendon by performing plantar flexion with maximum effort and consisted of 12 sets, each set including 15 s under tension; the sets were interspersed with 30 s rest periods. Three different loading types were used to simulate various conditions during sports: eccentric, isometric, and combined concentric/eccentric loading (Appendix A). The plantar flexion was performed with the knee flexed at 90° while lying down on the back. The range of motion in the angle for eccentric and concentric/eccentric loading was from 30° of plantar flexion to 15° of dorsiflexion; the isometric loading was performed at 0° (Appendix A) [24,25].

The simulation of the ADL included a five-minute session on an ergometer at 100 watts and 80 revolutions per minute, followed by 20 heel raises. Its aim was to simulate conditions typically preceding assessments in clinical and research practice, such as cycling or walking to the lab and climbing up the stairs [26].

### 2.7. Procedures

Approximately one week before the first wave of measurement, participants underwent a familiarization session. During this session, participants practiced the simulation of daily living (ADL) and sports activities (SP). Furthermore, the setup of the standardized position was recorded (height of the rack, toe-to-box and inter-metatarsal distances) so that the standardized position (Figure 2) could be replicated across all measurement sessions. Finally, participants were instructed to refrain from engaging in any physically strenuous activity 3 days prior to each wave of measurement.

For each of the three waves separated by 14 days (Figure 1), the participants followed the same procedure: at the start of the 1st visit, participants rested in the lying position for 5 min. Then, they assumed the predetermined seated position as established during the familiarization session, the points of measurement were marked, and the first set of standardized measurements (SM) was taken. After the first SM, the participants underwent the SP protocol simulating the sports activities (Appendix A). Then, immediately, they re-assumed the standardized position, and the second SM was taken. After 15 min of rest, the participants re-assumed the standardized position once more, and the third SM of the first visit was taken. The 2nd visit was conducted 24 h after the 1st visit and also began with a 5 min period of rest. Following the rest, participants assumed the standardized position, and the first SM was taken. After that, the simulation of ADL conducted. After completing the simulation, participants re-assumed the standardized position and the second SM of the 2nd visit was taken.

Sessions 1 to 3 were measured by Operator 1, while sessions 4 and 5 were measured by Operator 2 or 3, randomly assigned. All operators were doctoral students in kinesiology with prior training in identifying the distal and proximal Achilles tendon landmarks. Before data collection, they practiced marking and measuring at least 20 trials on pilot participants to ensure consistency in identification and device handling.

### 2.8. Statistical Analysis

The intraclass correlation coefficients (ICC) and their 95% confidence intervals were calculated using the irr package (version 0.84.1) in R statistical software. The analysis was conducted using the “two-way” model, “agreement” type, and “single” or “average” unit of analysis [27,28]. The magnitude of the intraclass correlation coefficient was interpreted based on its lower-bound of the 95% confidence interval (LCI), as follows: <0.50, poor reliability; 0.50 to 0.75, moderate reliability; 0.75 to 0.90, good reliability; and >0.90, excellent reliability [27].

We initially screened all the data for extreme outliers (values exceeding the mean ± 3SD), but none met our predefined exclusion criteria. To assess potential systematic bias between repeated measurements for immediate reliability, we compared exactly two measurements at a time (i.e., measurements 1 and 2, 2 and 3, and 1 and 3) using a Bland–Altman analysis (Appendix A). For each pair, the mean difference (bias) and 95% limits of agreement were calculated, and a paired t-test was performed to determine whether the bias was statistically significant. The magnitude of the observed bias was then compared to the minimal detectable change (MDC) to evaluate its clinical relevance.

The immediate reliability was calculated as a comparison between 3 single measurements within each SM for both legs. The short-term reliability was calculated as a comparison between 3 averages (of the 3 single measurements) from sessions 1, 2, and 3 only for an unloaded leg. The short-term reliability with simulation of activities of daily living was calculated as a comparison between 2 averages from sessions 4 and 5 (with the ADL protocol in between) for both legs. The short-term reliability with simulation of sport was calculated as a comparison between averages from session 1, 2, and 3 only for a loaded leg. The between-day reliability with and without simulation of sports was calculated as a comparison between 2 averages from sessions 1 and 4 for an unloaded and a loaded leg, respectively. The between-week reliability was calculated as a comparison between 2 averages from waves 1, 2, and 3 for sessions 1, 2, and 3 and only for an unloaded leg. The inter-rater reliability was calculated as a comparison between 2 averages from sessions 1 and 5 for an unloaded leg. The data used for the calculation of individual ICCs are summarized in Table 2.

The coefficient of variation (CV) was calculated by formula in R statistical software: (standard deviation/mean) × 100. The standard error of measurement (SEM) was calculated using plotrix package (version 3.8-4) package in R statistical software. The minimal detectable change (MDC) was calculated by formula in R statistical software: 1.96 × SEM × 2. All valid data points were analyzed, and no outliers were excluded.

## 3. Results

The median stiffness at the proximal point of the Achilles tendon was 852.5 N/m (IQR 194.5) and at the distal point, 1019.3 N/m (IQR 129.5). The reliability of the MyotonPRO measurement of Achilles tendon stiffness across different time frames and settings ranged from good to excellent (Table 3). The test-retest reliability, both with and without subject repositioning, was excellent with the lower bound of 95% confidence interval exceeding 0.9. The test-retest reliability, including effect of time (1 day to 2 weeks) and physical activity stimuli (ADLs, SPs), was good with the lower bound of 95% confidence interval exceeding 0.75 (Figure 3).

### Subgoup and Systemic Bias Analysis

In addition to the overall reliability analyses, we performed subgroup analyses comparing the two measurement sites on the Achilles tendon. The proximal measurement point yielded an intraclass correlation coefficient (ICC) of 0.966 (0.958–0.973), while the distal measurement point yielded an ICC of 0.956 (0.944–0.966) for immediate SM. These high and comparable ICC values at both sites further support the robustness and consistency of our standardized measurement protocol. Formal analysis of potential systematic bias was performed. The Bland–Altman analysis revealed statistically significant negative systematic biases ranging from −5.43 to −11.52 N/m (all *p* < 0.001) (Appendix A). However, in all cases, the bias remained within the range of minimal detectable change (MDC = ~15–~62 N/m), indicating that the observed differences are unlikely to be clinically meaningful.

**Figure 3 jfmk-10-00083-f003:**
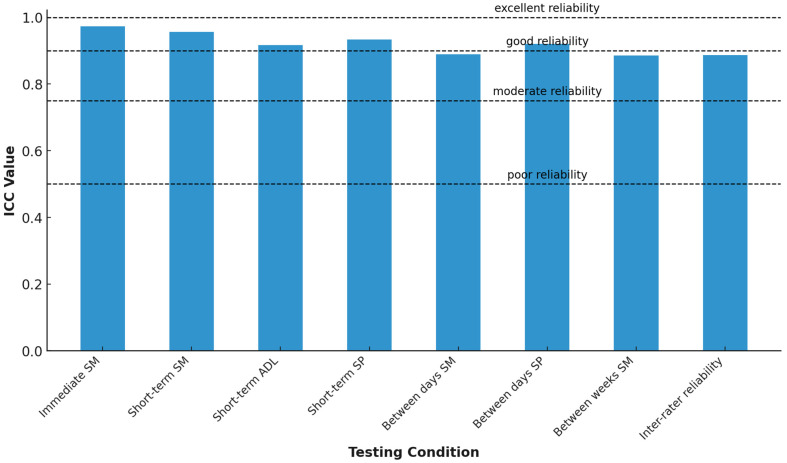
Graph of reliability analysis for different testing conditions.

## 4. Discussion

This study brings novel insights into the reliability of the MyotonPRO and provides standardized procedures for measuring Achilles tendon stiffness in clinical trials with patients. The main findings are that (1) the proposed standardized seating position is highly reliable in measuring Achilles tendon stiffness; (2) the standardized seating position allowed for highly reliable readings after repositioning the participant; (3) the simulated ADL did not influence Achilles tendon stiffness; (4) the simulated SP also did not influence Achilles tendon stiffness; (5) the records were consistent between days and weeks in all testing conditions. However, a mild decrease in reliability was noted with longer testing intervals, potentially reflecting variations in tendon biomechanics over the longer periods (weeks). Collectively, these findings recommend following the proposed standardized procedures to measure Achilles tendon stiffness with the MyotonPRO, ensuring high reliability regardless of the time between measurements and physical activity.

Our additional subgroup analyses indicated that the proximal and distal measurement points provide similarly high reliability, with ICCs of 0.966 and 0.956, respectively. This finding confirms that both regions of the Achilles tendon can be reliably measured using the MyotonPRO in the standardized seated position. Although we did not perform formal sensitivity analyses, these subgroup results, in conjunction with our other reliability assessments, strongly support the validity and robustness of our measurement approach. We acknowledge that future studies may benefit from incorporating sensitivity analyses to further explore potential variability across different analytical models.

Previous studies have tested the MyotonPRO inter-session reliabilities (up to one week apart) for Achilles tendon stiffness in a relaxed prone, ankle-free position (i.e., with the foot hanging freely), finding ICCs from 0.80 to 0.90, SEMs from 24.7 to 58.8 N/m and MDCs from 58.8 to 69.0 N/m [17,29,30]. These outcomes can be improved by using a splint to maintain the same ankle range of motion, resulting in ICCs of 0.95, SEMs of 13.8 N/m and MDCs of 36.7 N/m [15]. Building on this, our study designed a standardized seated position and provided a comprehensive reliability report under different exercise stimuli and time intervals. Results showed comparable excellent outcomes in all settings (ICC from 0.89 to 0.97, SEM from 4.0 to 15.9 N/m, MDC from 15.5 to 62.4 N/m), thus recommending its implementation in future studies involving Achilles tendon stiffness measurement.

In comparison with other techniques, such as shear-wave ultrasound elastography (SWUE) or ultrasound-based displacement measurements during maximal voluntary contraction, the MyotonPRO offers several advantages. Most importantly, it is less operator-dependent, providing direct quantitative results for stiffness without the need for extensive ultrasound expertise or visual interpretation of tissue images. While SWUE can provide detailed tendon structure, it requires specialized training and reproducibility can vary widely. In contrast, our results suggest that the MyotonPRO provides a convenient and portable solution, particularly in situations where visual imaging is not essential but reliable numerical estimates of tendon stiffness are required. Nevertheless, future studies could compare seated measurements to prone or standing methods to determine whether the seated position confers additional benefits in terms of participant comfort, reduced muscle tension, or ease of landmark identification.

Not only does this study show superior reliability outcomes with a standardized seated position, it also demonstrated consistent outcomes between days and weeks. Although the inter-session reliability has been previously assessed at 7 days [17,29,30] and 5 days apart [15], to our knowledge, our study is the first to assess reliability over longer time intervals (14 days). Measurements between weeks are crucial for evaluating the clinical effect in tendinopathy patients, where initial improvements may take weeks or even months to occur [16,31]. Nonetheless, considering magnitude of errors in relation to the absolute tendon stiffness (i.e., 850 to 1000 N/m), the expected variability across days would represent less than 0.35% of the outcome, which is highly acceptable.

Muscle tension may also influence stiffness measurements, as any residual activation in the triceps surae could slightly alter the measured tendon properties. By seating participants with knees bent at 90° and encouraging full relaxation, we attempted to minimize this confounder. However, more sensitive electromyographic (EMG) monitoring might be employed in future work to ensure negligible muscle activation during testing. Additionally, our results revealed that both activities of daily living (ADL) and the simulation of sport had no notable immediate impact on Achilles tendon stiffness. While this suggests a certain resilience of tendon stiffness to single bouts of loading, it also provides practical insight for planning measurement schedules in clinical or research settings. Nonetheless, the cumulative effects of repetitive or prolonged loading over days or weeks remain an important area for further investigation.

The Achilles tendon stiffness showed no apparent changes after both exercise stimuli. Accordingly, researchers can expect that a low-to-moderate physical activity, such as walking or cycling, will have no effect on Achilles tendon stiffness. Similarly, a single high-impact effort (i.e., isometric maximal plantar flexion) seems not to evoke noticeable changes in the records. Because this is the first report detailing measurement errors under these specific exercise conditions, future research could expand upon this work by examining other forms of loading (e.g., plyometric jumps or sprint protocols) and longer bouts of repetitive stress. Future studies are encouraged to determine the reliability of Achilles tendon stiffness measurement under other stimuli (e.g., jumps or sprints). This information is essential for designing high-quality clinical trials capable of identifying real changes after a given intervention. All in all, the cumulative effect of loading during days or weeks should be considered by researchers, as tendon tissue can react after consistent loading over weeks to months [24].

Regarding clinical application, the minimal detectable change (MDC) values observed (ranging from ~15 to ~62 N/m) provide a frame of reference for interpreting “true” changes in tendon stiffness. For instance, if a treatment yields a stiffness alteration below the MDC, clinicians might question whether this shift is clinically meaningful or simply a random error. Thus, the absolute indices reported here serve as practical thresholds when monitoring rehabilitation or training outcomes.

In sum, this study provides background for assessing Achilles tendon stiffness in clinical studies and explains what to expect from a reliability standpoint when measuring with intervals of minutes, days, and weeks between measurements. The information herein lays the foundation for determining accurate, meaningful changes in Achilles tendon stiffness after a rehabilitation or training program. Based on the data, it is recommended to use a standardized position for each measurement to achieve excellent reliability of MyotonPRO in the Achilles tendon region over a time horizon between measurements ranging from 10–15 min to 2 weeks. The sitting position may be the best option as it causes less of a stretch in the plantar flexors compared to lying on the belly with a 90° angle in the ankle. This means that the tendon tissue is less affected by the muscle during measurements. Standardizing the ankle position, relaxing the plantar flexors, and fixing the position of MyotonPRO proved to be crucial in maintaining reliability, even with several weeks between measurements.

There are some limitations to this study that should be noted. Only healthy participants were recruited, so the results may be different in tendinopathy patients. This choice may introduce biases, as the stiffness, and any potential central inhibition in clinical populations could differ from those in healthy individuals. The stiffness measurement was only taken at two points, so there may be different results for different points on the Achilles tendon. However, we did not observe any significant differences between the proximal and distal locations in data analysis. The simulation of sport was conducted on a dynamometer, which may be different in comparison to real sport conditions. However, the dynamometer enabled a comparable and controlled environment for all participants. The ADL simulation involved an ergometer followed by heel rises, so the results for actual walking or stairs climbing may differ. However, this setup again provided a comparable and controlled environment.

## 5. Conclusions

Overall, the MyotonPRO has demonstrated good to excellent reliability when utilizing a standardized participant position and precise positioning of the measuring device on the rack to eliminate any deviations in the angle relative to the measured Achilles tendon. Nevertheless, these findings primarily apply to healthy young adults under controlled conditions, and must be interpreted with caution in clinical scenarios involving older or symptomatic populations. The reliability of the MyotonPRO has been confirmed for the same operator across different days and weeks, as well as between two operators using the same standardized measurement protocol. Within these parameters, the MyotonPRO can be confidently employed to capture tendon stiffness values in a consistent manner. However, additional research is warranted to determine the device’s performance in diverse patient groups and real-world clinical settings.

In conclusion, while these findings underscore the reliability of MyotonPRO measurements for Achilles tendon stiffness in healthy young adults, they should be interpreted with caution when applying them to clinical populations. Future work might explore other sampling sites, employ alternative exercise stimuli, and include symptomatic individuals to better elucidate the MyotonPRO’s full potential in rehabilitation and performance contexts.

## Figures and Tables

**Figure 1 jfmk-10-00083-f001:**
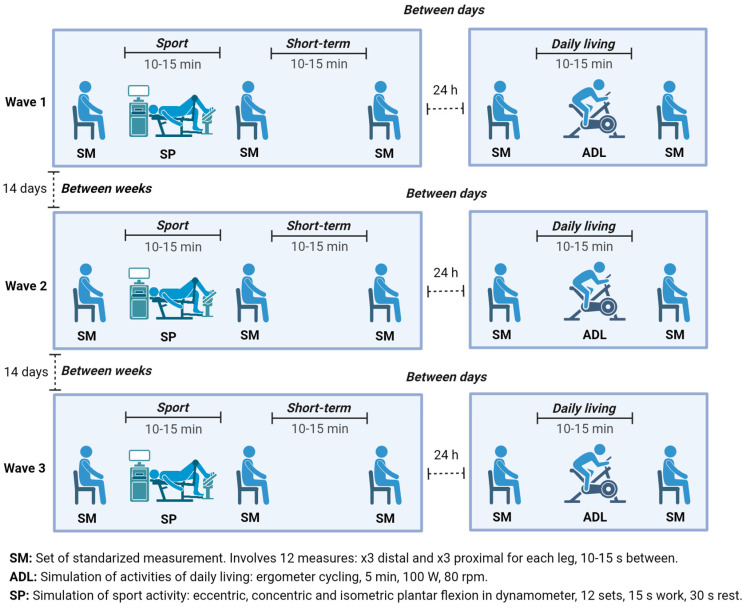
Scheme of measurements-time intervals and different settings.

**Figure 2 jfmk-10-00083-f002:**
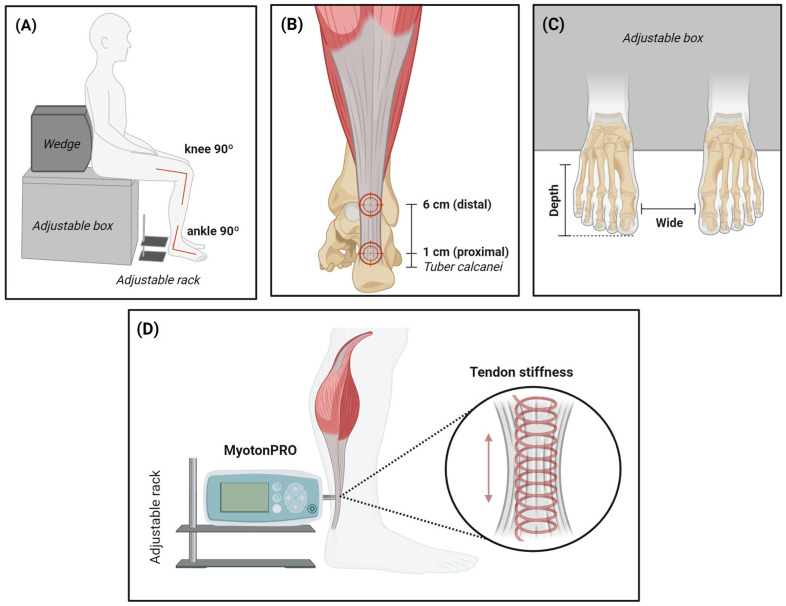
Standardized measurement of Achilles tendon stiffness. (**A**) Sitting position; (**B**) Measured sites; (**C**) Foot positioning; (**D**) MyotonPRO positioning.

**Table 1 jfmk-10-00083-t001:** Overview of time frames, repositioning, and stimuli.

	Time Interval Between Measurements	In-Between Repositioning	In-Between Stimulus	Purpose
Immediate SM	10–15 s	NO	NO	Consistency of measurement
Short-term SM	10–15 min	YES	NO	Test-retest reliability
Short-term ADL	10–15 min	YES	ADL	Impact of low-intensity loading
Short-term SP	10–15 min	YES	SP	Impact of high-intensity loading
Between days SM	24 h	YES	NO	Test-retest reliability between days
Between days SP	24 h	YES	SP	Impact of high-intensity loading
Between weeks SM	14 days	YES	NO	Test-retest reliability between weeks
Inter-rater reliability	24 h	YES	NO	Operator influence on reliability

SM: Set of standardized measurement. Involves 12 measures: ×3 distal and ×3 proximal for each leg, 10–15 s between. ADL: Simulation of activities of daily living: ergometer cycling, 5 min, 100 W, 80 rpm. SP: Simulation of sport activity: eccentric, concentric, and isometric plantar flexion in dynamometer, 12 sets, 15 s work, 30 s rest.

**Table 2 jfmk-10-00083-t002:** Data used for calculation of the intraclass correlation coefficients.

	Number of Measurements Compared	Unit of Analysis	Sessions Included	Legs Included	Total Number of Measurements Included
Immediate SM	3	single	1 to 5	both	1440
Short-term SM	3	mean	1 to 3	unloaded	144
Short-term ADL	2	mean	4 and 5	both	192
Short-term SP	3	mean	1 to 3	loaded	144
Between days SM	2	mean	1 and 4	unloaded	96
Between days SP	2	mean	1 and 4	loaded	96
Between weeks SM	2	mean	1 and 2	unloaded	128
Inter-rater reliability	2	mean	1 and 5	unloaded	96

SM: Set of standardized measurement. Involves 12 measures: ×3 distal and ×3 proximal for each leg, 10–15 s between. ADL: Simulation of activities of daily living: ergometer cycling, 5 min, 100 W, 80 rpm. SP: Simulation of sport activity: eccentric, concentric, and isometric plantar flexion in dynamometer, 12 sets, 15 s work, 30 s rest.

**Table 3 jfmk-10-00083-t003:** Results from reliability analyses for the different testing conditions.

Testing Condition	ICC (95% CI)	CV (%)	SEM (N/m)	MDC (N/m)	Median (IQR) (N/m)
Immediate SM	0.973 (0.968 to 0.978)	15.99	3.95	15.46	951 (238.25)
Short-term SM	0.956 (0.929 to 0.974)	16.14	12.50	49.01	948 (232)
Short-term ADL	0.917 (0.875 to 0.945)	16.00	10.88	42.63	955.5 (236.5)
Short-term SP	0.933 (0.891 to 0.96)	15.74	12.26	48.07	944 (237.5)
Between days SM	0.889 (0.802 to 0.938)	16.51	15.65	61.34	945.5 (254.75)
Between days SP	0.920 (0.859 to 0.955)	16.67	15.91	62.35	956.5 (234.75)
Between weeks SM	0.886 (0.811 to 0.931)	15.82	13.05	51.17	953 (219)
Inter-rater reliability	0.887 (0.798 to 0.937)	16.48	15.61	61.21	942 (252.5)

ICC: Intraclass correlation coefficient. CV: Coefficient of variation. SEM: Standard error of measurement. MDC: Minimal detectable change.

## Data Availability

The data presented in this study are available on request from the corresponding author due to the need to protect participant privacy and confidentiality, as stipulated by ethical guidelines and data protection regulations.

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
