# Peer review of "Assessing the Test-Retest Reliability of MyotonPRO for Measuring Achilles Tendon Stiffness"

_jfmk, 2025, doi:10.3390/jfmk10010083_

Round 1
Reviewer 1 Report
Comments and Suggestions for Authors
Dear corresponding Author,
thanks for submitting your paper.
Please check the names because there is a last name that is double.
Please delete the numbers into the keywords space.
Please check the format of the references into the text because it is not in line with the Authors' instructions.
Please check the font in the paper, in some parts it is different.
The introduction lacks of the study hypothesis. What is the hypothesis of the study?
In the experimental approach one thing is not clear to me. Why 10-15 s and 10-15 minutes? If the ragne 10-15s can be understood and real, it is hard to me to understand why 5 minutes of range. Did you perform the retest 10 or 15 minute after? Please explain it better in the text. It is crucial in case of someone wants to replicate the study.
Please check the title of 2.5 section.
I have some doubt if the motor protocols that you used can be considered as a simulation of sport and daily activities. Do you have some references?
In the limitations section, you highlighted a crucial point: the study was conducted exclusively on healthy subjects, and it remains uncertain what might occur in individuals with tendinopathy. Well done. However, I believe it is very important to include this point in the conclusions as well. It should be emphasized that the results are specific to healthy subjects and that further studies, similar to this one, need to be conducted on individuals with Achilles tendon issues.
The study is well-conducted, and I have no methodological comments to make.
Author Response
Response to Reviewer 1 Comments
|
JFMK-3411868
Assessing The Test-Retest Reliability of MyotonPRO for Measuring Achilles Tendon Stiffness Journal of Functional Morphology and Kinesiology
Reviewer #1
Comments 1:
Dear corresponding Author,
thanks for submitting your paper.
Response 1:
Thank you for your positive feedback. We appreciate your constructive suggestions.
Comments 2:
Please check the names because there is a last name that is double.
Please delete the numbers into the keywords space.
Please check the format of the references into the text because it is not in line with the Authors' instructions.
Please check the font in the paper, in some parts it is different.
Response 2:
Thank you for your suggestions, we fixed all issues you mentioned.
Comments 3:
The introduction lacks of the study hypothesis. What is the hypothesis of the study?
Response 3:
Thank you for pointing this out. We have now explicitly stated our main hypothesis at the end of the Introduction section. Specifically, we hypothesize that the MyotonPRO device will demonstrate good to excellent test-retest reliability in measuring Achilles tendon stiffness over various time frames (seconds, minutes, days, and weeks) and in response to different activity stimuli (activities of daily living and sport).
Line 181 “We hypothesized that, under standardized conditions, the MyotonPRO would demonstrate good to excellent test-retest reliability (intraclass correlation coefficient > 0.75) across all assessed intervals and activity stimuli.
Regarding the time frames specifically, immediate measurements at 10–15 s apart allowed us to capture consecutive trials without any repositioning of the participant. The 10–15 min interval served as a short-term retest period, encompassing practical requirements such as repositioning participants, marking the measurement site, and, in some cases, performing a quick activity stimulus or rest. This slight variability (i.e., sometimes closer to 10 min, sometimes 15 min) reflects realistic conditions in clinical and research settings, where minor procedural or participant-related delays commonly occur.
By providing both a clear hypothesis and a rationale for the time intervals used, this study addresses a gap in the literature and offers a reproducible methodology for reliably assessing Achilles tendon stiffness in healthy and potentially clinical populations.”
Comments 4:
In the experimental approach one thing is not clear to me. Why 10-15 s and 10-15 minutes? If the ragne 10-15s can be understood and real, it is hard to me to understand why 5 minutes of range. Did you perform the retest 10 or 15 minute after? Please explain it better in the text. It is crucial in case of someone wants to replicate the study.
Response 4:
We acknowledge that our explanation of the 10–15 min interval was not entirely clear in the original text. The short interval (10–15 s) refers to the time needed to collect consecutive measurements without repositioning the participant - essentially immediate measurements. The 10–15 min interval was used to represent short-term retesting after repositioning and any brief required procedures (e.g., marking sites, allowing the participant to get up from and return to the measurement station, and other practical logistical factors). In practice, this short-term interval sometimes took closer to 10 min, whereas in other cases it extended to about 15 min. This slight variation in timing aligns with real-world clinical or research settings, in which minor delays are common. We have now clarified this point in the revised Introduction to facilitate replication.
Line 246 “2.3. Rationale for time intervals
The 10–15 s interval enabled immediate consecutive measurements without repositioning, reflecting direct repeatability. The 10–15 min interval accounted for practical short-term repositioning and minor procedural tasks. The 24 h interval represented next-day follow-up, commonly employed in clinical check-ups. Finally, the 14-day interval allowed the assessment of stability over a longer period, relevant for monitoring rehabilitation programs.”
Comments 5:
Please check the title of 2.5 section.
Response 5:
Thank you for your suggestion, we fixed the issue you mentioned.
Comments 6:
I have some doubt if the motor protocols that you used can be considered as a simulation of sport and daily activities. Do you have some references?
Response 6:
We understand your doubts. Our goal was to design practical, reproducible protocols that approximate the loading conditions experienced by the Achilles tendon during typical daily movements and sport-like activities in a controlled laboratory environment. For ADL, we combined a short period of moderate-intensity cycling (100 W, 80 rpm), relevant for active commuting, with heel raises, which are commonly used to simulate walking [26]. For the sport simulation, we employed repeated maximal eccentric, isometric, and concentric plantar flexion on an isokinetic dynamometer, a recognized approach for replicating high-intensity stresses on the Achilles tendon [24, 25]. While these protocols may not capture every nuance of real-world activity, they provide a standardized, lab-based approximation of both lower-intensity daily tasks and vigorous loading conditions.
- Kubo K, Ikebukuro T, Maki A, Yata H, Tsunoda N. Time course of changes in the human Achilles tendon properties and metabolism during training and detraining in vivo. European Journal of Applied Physiology. 2012 Jul;112(7):2679–91. DOI: 10.1007/s00421-011-2248-x
- Kelly, M. Al-Uzri, and S. O’Neill, “49 Effect Of Eccentric Training On Isokinetic Endurance Of Calf With Reliability Testing,” Br J Sports Med, vol. 48, no. Suppl 2, p. A32, Sep. 2014, doi: 10.1136/bjsports-2014-094114.49.
Comments 7:
In the limitations section, you highlighted a crucial point: the study was conducted exclusively on healthy subjects, and it remains uncertain what might occur in individuals with tendinopathy. Well done. However, I believe it is very important to include this point in the conclusions as well. It should be emphasized that the results are specific to healthy subjects and that further studies, similar to this one, need to be conducted on individuals with Achilles tendon issues.
Response 7:
Thank you for this suggestion. We agree it is important to reiterate that the findings may not translate directly to populations with Achilles tendinopathy. We have now added a clarifying statement in the „Discussion“ section for the need for further research in clinical populations.
Line 567 „Nevertheless, these findings primarily apply to healthy young adults under controlled conditions, and must be interpreted with caution in clinical scenarios involving older or symptomatic populations.“
Comments 8:
The study is well-conducted, and I have no methodological comments to make.

Reviewer 2 Report
Comments and Suggestions for Authors
This manuscript examines the reliability of the MyotonPRO device for measuring Achilles tendon stiffness across various time intervals and conditions. The study addresses a relevant topic in clinical biomechanics, but there are several methodological concerns and limitations that should be addressed. Additionally, the absence of line numbers makes the review process more challenging.
Introduction
1. Paragraph 2: The authors should provide clearer and more specific references to support their statements.
2. Last Paragraph: The study hypothesis is missing and should be explicitly stated.
3. Rationale for MyotonPRO: The justification for selecting the MyotonPRO over alternative technologies is insufficiently explained. Please expand on the rationale, emphasizing the advantages or unique features of the device.
Materials and Methods
1. It is recommended that the authors describe the sample characteristics first before detailing the experimental approach.
2. On page 2, in the last sentence, terms like "ADL" and "SP" should be spelled out in the text, even if they are defined in a figure.
3. Figure 2: Including actual photographs of the measurement setup alongside the hand-drawn diagrams would enhance clarity and understanding.
4. The sample size (n=8) is very small, despite the authors' use of Zou’s formula for justification. The introduction mentions the unsuitability of this approach for tendinopathy cases due to central inhibition of plantar flexors. However, only healthy participants were recruited, creating a contradiction. Please clarify this discrepancy and acknowledge it as a limitation. Age Range: The restricted age range (median age: 27.5 years) limits the generalizability of findings. Consider discussing how this age range compares to the typical demographic of tendinopathy patients.
5. Specify the model of the MyotonPRO used in the study. Provide more details about the positioning of the adjustable rack for standardization. If possible, include a power analysis for the various reliability measures tested. Reporting an overall average may also be helpful.
6. In subsection 2.6, the expertise and training of the three operators should be described, particularly regarding their proficiency in identifying anatomical landmarks.
7. The choice of specific time intervals (10–15s, 10–15min, 24h, 14 days) is not adequately justified. Please explain the rationale for these intervals.
8. The handling of outliers is not mentioned in the methodology. This should be addressed.
Results
1. The results suggest a decline in reliability with longer intervals. Possible causes should be elaborated upon in the Discussion section.
2. There is no analysis of potential systematic bias between measurements. Consider including this analysis.
Discussion
1. Comparisons with other measurement techniques should be made to highlight the relative strengths or limitations of the MyotonPRO.
2. The impact of muscle tension on stiffness measurements is insufficiently discussed and warrants more attention.
3. The clinical relevance of absolute indices (e.g., MDC, N/m) should be clearly described.
Conclusion
The statement, “These findings suggest that the MyotonPRO can be confidently employed in clinical studies and interventions, providing accurate and consistent measurements of Achilles tendon stiffness,” appears overly optimistic given the study’s limitations. Please temper this conclusion and clarify the specific contexts in which the device may be reliably applied.
Author Response
Response to Reviewer 2 Comments
|
JFMK-3411868
Assessing The Test-Retest Reliability of MyotonPRO for Measuring Achilles Tendon Stiffness Journal of Functional Morphology and Kinesiology
Reviewer #2
Comments 1:
This manuscript examines the reliability of the MyotonPRO device for measuring Achilles tendon stiffness across various time intervals and conditions. The study addresses a relevant topic in clinical biomechanics, but there are several methodological concerns and limitations that should be addressed. Additionally, the absence of line numbers makes the review process more challenging.
Response 1:
Thank you for your feedback. We appreciate your constructive suggestions.
Introduction
Comments 2:
Paragraph 2: The authors should provide clearer and more specific references to support their statements.
Response 2:
We have added an additional reference in Paragraph 2 to specifically support claims regarding the magnitude of Achilles tendon loads, the strain thresholds for tendon damage, and the link between excessive elongation and tendon ruptures.
- H.-C. Wang, “Mechanobiology of tendon,” J Biomech, vol. 39, no. 9, pp. 1563–1582, 2006. DOI: 10.1016/j.jbiomech.2005.05.011
Comments 3:
Last Paragraph: The study hypothesis is missing and should be explicitly stated.
Response 3:
We have updated the final paragraph of the Introduction to clearly and explicitly state our hypothesis that, under standardized conditions, the MyotonPRO would demonstrate good to excellent test-retest reliability (ICC > 0.75) across all assessed intervals and activity stimuli.
Line 181 “We hypothesized that, under standardized conditions, the MyotonPRO would demonstrate good to excellent test-retest reliability (intraclass correlation coefficient > 0.75) across all assessed intervals and activity stimuli.
Regarding the time frames specifically, immediate measurements at 10–15 s apart allowed us to capture consecutive trials without any repositioning of the participant. The 10–15 min interval served as a short-term retest period, encompassing practical requirements such as repositioning participants, marking the measurement site, and, in some cases, performing a quick activity stimulus or rest. This slight variability (i.e., sometimes closer to 10 min, sometimes 15 min) reflects realistic conditions in clinical and research settings, where minor procedural or participant-related delays commonly occur.”
By providing both a clear hypothesis and a rationale for the time intervals used, this study addresses a gap in the literature and offers a reproducible methodology for reliably assessing Achilles tendon stiffness in healthy and potentially clinical populations.
Comments 4:
Rationale for MyotonPRO: The justification for selecting the MyotonPRO over alternative technologies is insufficiently explained. Please expand on the rationale, emphasizing the advantages or unique features of the device.
Response 4:
We have expanded the discussion of the MyotonPRO, highlighting its key advantages—such as portability, user-friendly interface, minimal operator dependency compared to ultrasound elastography, and its ability to quickly and reliably measure soft tissue stiffness.
Line 139 “Compared to more operator-dependent modalities (e.g., ultrasound elastography), the MyotonPRO requires less specialized training and reduces user-dependent variability. Its portability and straightforward setup make it a convenient tool for researchers and clinicians to measure soft-tissue stiffness quickly and accurately“.
Materials and Methods
Comments 5:
It is recommended that the authors describe the sample characteristics first before detailing the experimental approach.
Response 5:
We have moved the “Sample” section above the “Experimental approach” to present participant characteristics and eligibility criteria prior to the study design.
Comments 6:
On page 2, in the last sentence, terms like "ADL" and "SP" should be spelled out in the text, even if they are defined in a figure.
Response 6:
We have spelled out “activities of daily living (ADL)” and “sport protocol (SP)” in the main text, even though they appear in Table 1 and Figure captions.
Comments 7:
Figure 2: Including actual photographs of the measurement setup alongside the hand-drawn diagrams would enhance clarity and understanding.
Response 7:
We thank for suggesting the inclusion of actual photographs to improve clarity. We have now added a “Supplementary Material” file containing an actual photo of the adjustable rack and the MyotonPRO. In Figure 2, we have included an infographic that comprehensively illustrates the standardized measurement setup and procedure. This infographic details the specific positioning of the participant, the placement of the measuring device, and relevant anatomical landmarks. By visually outlining each step, we provide a clear, replicable protocol for use in both research and practice. Moreover, we believe that our approach—offering a standardized protocol rather than merely reporting numerical data—goes beyond many prior studies that focus exclusively on measurement outcomes without specifying a reproducible methodology. Our goal is to ensure that future researchers and clinicians can precisely replicate our methods, thereby enhancing the consistency and reliability of Achilles tendon stiffness assessments using the MyotonPRO.
Note: Added to “Supplementary material”
Comments 8:
The sample size (n=8) is very small, despite the authors' use of Zou’s formula for justification. The introduction mentions the unsuitability of this approach for tendinopathy cases due to central inhibition of plantar flexors. However, only healthy participants were recruited, creating a contradiction. Please clarify this discrepancy and acknowledge it as a limitation. Age Range: The restricted age range (median age: 27.5 years) limits the generalizability of findings. Consider discussing how this age range compares to the typical demographic of tendinopathy patients.
Response 8:
We acknowledge the small sample size and the use of healthy participants as a limitation. We have clarified that although the MyotonPRO may be suitable for tendinopathy, we only tested healthy individuals here. The age range is also addressed as a limitation regarding generalizability.
212 „These criteria yielded a relatively narrow age range (median: 27.5 years), which may limit generalizability to a broader population. Furthermore, although the MyotonPRO may be used in tendinopathy cases, our sample included only healthy participants.”
Line 599 „Nevertheless, these findings primarily apply to healthy young adults under controlled conditions, and must be interpreted with caution in clinical scenarios involving older or symptomatic populations.“
Comments 9:
Specify the model of the MyotonPRO used in the study. Provide more details about the positioning of the adjustable rack for standardization. If possible, include a power analysis for the various reliability measures tested. Reporting an overall average may also be helpful.
Response 9:
We have specified the exact MyotonPRO model (MyotonPRO, Muomeetria, Tallinn, Estonia, Model 000607). We have now added a “Supplementary Material” file containing an actual photo of the adjustable rack and the MyotonPRO. We have also clarified that we relied on Zou’s formula to determine the sample size for ICC, but due to the scope of this reliability study, we did not perform a separate power analysis for each subset measure.
Comments 10:
In subsection 2.6, the expertise and training of the three operators should be described, particularly regarding their proficiency in identifying anatomical landmarks.
Response 10:
We have now described the training levels and background of the three operators, emphasizing their experience in locating and marking anatomical landmarks.
Line 364 „Sessions 1 to 3 were measured by Operator 1, while sessions 4 and 5 were randomly assigned to Operator 2 or 3. All operators were doctoral students in kinesiology with prior training in identifying the distal and proximal Achilles tendon landmarks. Before data collection, they practiced marking and measuring at least 20 trials on pilot participants to ensure consistency in identification and device handling.“
Comments 11:
The choice of specific time intervals (10–15s, 10–15min, 24h, 14 days) is not adequately justified. Please explain the rationale for these intervals.
Response 11:
We have expanded our explanation in section 2., noting practical, clinical, and research considerations that led us to choose these intervals, such as immediate consecutive measures, short-term repositioning, next-day follow-up, and a more extended follow-up.
Line 246 „2.3. Rationale for time intervals
The 10–15 s interval enabled immediate consecutive measurements without repositioning, reflecting direct repeatability. The 10–15 min interval accounted for practical short-term repositioning and minor procedural tasks. The 24 h interval represented next-day follow-up, commonly employed in clinical check-ups. Finally, the 14-day interval allowed the assessment of stability over a longer period, relevant for monitoring rehabilitation programs.“
Comments 12:
The handling of outliers is not mentioned in the methodology. This should be addressed.
Response 12:
We have clarified that we assessed all collected data points for extreme values, and no outliers were removed. We included all measurements in the statistical analysis.
Line 377 “We initially screened all data for extreme outliers (values exceeding the mean ± 3SD), but none met our predefined exclusion criteria. To assess potential systematic bias between repeated measurements for immediate reliability, we compared exactly two measurements at a time (i.e., measurements 1 and 2, 2 and 3, and 1 and 3) using a Bland–Altman analysis (Supplementary Material). For each pair, the mean difference (bias) and 95% limits of agreement were calculated, and a paired t-test was performed to determine whether the bias was statistically significant. The magnitude of the observed bias was then compared to the minimal detectable change (MDC) to evaluate its clinical relevance.”
Results
Comments 13:
The results suggest a decline in reliability with longer intervals. Possible causes should be elaborated upon in the Discussion section.
Response 13:
We have added Figure 3. Graph of reliability analysis for different testing conditions.
and brief discussion on potential reasons (e.g., normal daily fluctuations in tendon properties, variations in participant muscle tension) for the decreased ICC values over longer intervals (in revised Discussion).
Line 454 „However, a mild decrease in reliability was noted with longer testing intervals, potentially reflecting variations in tendon biomechanics over the longer periods (weeks).“
Comments 14:
There is no analysis of potential systematic bias between measurements. Consider including this analysis.
Response 14:
We appreciate the comment regarding the analysis of potential systematic bias. In response, we have incorporated a Bland–Altman analysis into our study to evaluate any systematic bias between repeated measurements. The Bland–Altman plots and corresponding statistics are now included in the Methods and Results sections (see Supplementary Material – Systematic Bias Analysis).
Line 434 “Formal analysis of potential systematic bias was performed. The Bland-Altman analysis revealed statistically significant negative systematic biases ranging from -5.43 to -11.52 N/m (all p < 0.001) (Supplementary material). However, in all cases, the bias remained within the range of minimal detectable change (MDC = ~15 – ~62 N/m), indicating that the observed differences are unlikely to be clinically meaningful.”
Discussion
Comments 15:
Comparisons with other measurement techniques should be made to highlight the relative strengths or limitations of the MyotonPRO.
Response 15:
We have expanded the comparison with ultrasound elastography and other relevant methods, emphasizing the MyotonPRO’s strengths and potential limitations.
Line 478 “In comparison with other techniques, such as shear-wave ultrasound elastography (SWUE) or ultrasound-based displacement measurements during maximal voluntary contraction, the MyotonPRO offers several advantages. Most importantly, it is less operator-dependent, providing direct quantitative results for stiffness without the need for extensive ultrasound expertise or visual interpretation of tissue images. While SWUE can provide detailed tendon structure, it requires specialized training and reproducibility can vary widely. In contrast, our results suggest that the MyotonPRO provides a convenient and portable solution, particularly in situations where visual imaging is not essential but reliable numerical estimates of tendon stiffness are required. Nevertheless, future studies could compare seated measurements to prone or standing methods to determine whether the seated position confers additional benefits in terms of participant comfort, reduced muscle tension, or ease of landmark identification.”
Comments 16:
The impact of muscle tension on stiffness measurements is insufficiently discussed and warrants more attention.
Response 16:
We have added a section explaining how residual muscle tension may affect Achilles tendon stiffness measurements and how our standardized seated protocol aimed to minimize this effect.
Line 502 „Muscle tension may also influence stiffness measurements, as any residual activation in the triceps surae could slightly alter the measured tendon properties. By seating participants with knees bent at 90° and encouraging full relaxation, we attempted to minimize this confounder. However, more sensitive electromyographic (EMG) monitoring might be employed in future work to ensure negligible muscle activation during testing. Additionally, our results revealed that both activities of daily living (ADL) and the simulation of sport had no notable immediate impact on Achilles tendon stiffness. While this suggests a certain resilience of tendon stiffness to single bouts of loading, it also provides practical insight for planning measurement schedules in clinical or research settings. Nonetheless, the cumulative effects of repetitive or prolonged loading over days or weeks remain an important area for further investigation.”
Comments 17:
The clinical relevance of absolute indices (e.g., MDC, N/m) should be clearly described.
Response 17:
We have clarified how the minimal detectable change (MDC) and absolute stiffness values (in N/m) could inform clinicians or researchers about the threshold for meaningful differences in tendon stiffness.
Line 526 “Regarding clinical application, the minimal detectable change (MDC) values observed (ranging from ~15 N/m to ~62 N/m) provide a frame of reference for interpreting “true” changes in tendon stiffness. For instance, if a treatment yields a stiffness alteration below the MDC, clinicians might question whether this shift is clinically meaningful or simply a random error. Thus, the absolute indices reported here serve as practical thresholds when monitoring rehabilitation or training outcomes.”
Conclusion
Comments 18:
The statement, “These findings suggest that the MyotonPRO can be confidently employed in clinical studies and interventions, providing accurate and consistent measurements of Achilles tendon stiffness,” appears overly optimistic given the study’s limitations. Please temper this conclusion and clarify the specific contexts in which the device may be reliably applied.
Response 18:
We have revised the conclusion to include caveats, noting that our findings are primarily applicable to healthy young adults in controlled testing conditions, and caution must be exercised when extrapolating to broader or clinical populations.
599 „Nevertheless, these findings primarily apply to healthy young adults under controlled conditions, and must be interpreted with caution in clinical scenarios involving older or symptomatic populations.“
Line 575 „In conclusion, while these findings underscore the reliability of MyotonPRO measurements for Achilles tendon stiffness in healthy young adults, they should be interpreted with caution when applying them to clinical populations. Future work might explore other sampling sites, employ alternative exercise stimuli, and include symptomatic individuals to better elucidate the MyotonPRO’s full potential in rehabilitation and performance contexts.”

Reviewer 3 Report
Comments and Suggestions for Authors
Dear authors,
Your manuscript present novel insights into the reliability of the MyotonPRO for Achilles tendon stiffness. Please allow me to share some of my thoughts, that could probably help you:
- Clearly state the hypotheses being tested in the introduction.
- Provide a detailed rationale for the sample size and explain its implications for reliability studies.
- Include more robust statistical analysis to account for variability and potential outliers.
- Expand the discussion of the effects of ADL and SP on Achilles tendon stiffness, linking it to broader clinical applications.
- Revise the limitations section to acknowledge potential biases introduced by using healthy participants.
- Improve visual data representation, adding plots or graphs to complement the tables.
- Consider performing subgroup analyses or sensitivity analyses to strengthen the findings.
Thank You
Author Response
Response to Reviewer 3 Comments
|
JFMK-3411868
Assessing The Test-Retest Reliability of MyotonPRO for Measuring Achilles Tendon Stiffness Journal of Functional Morphology and Kinesiology
Reviewer #3
Dear authors,
Your manuscript present novel insights into the reliability of the MyotonPRO for Achilles tendon stiffness. Please allow me to share some of my thoughts, that could probably help you:
Comments 1:
Clearly state the hypotheses being tested in the introduction.
Response 1:
Thank you for pointing this out. We have now explicitly stated our main hypothesis at the end of the Introduction section. Specifically, we hypothesize that the MyotonPRO device will demonstrate good to excellent test-retest reliability in measuring Achilles tendon stiffness over various time frames (seconds, minutes, days, and weeks) and in response to different activity stimuli (activities of daily living and sport).
Line 181 „We hypothesized that, under standardized conditions, the MyotonPRO would demonstrate good to excellent test-retest reliability (intraclass correlation coefficient > 0.75) across all assessed intervals and activity stimuli.
Regarding the time frames specifically, immediate measurements at 10–15 s apart allowed us to capture consecutive trials without any repositioning of the participant. The 10–15 min interval served as a short-term retest period, encompassing practical requirements such as repositioning participants, marking the measurement site, and, in some cases, performing a quick activity stimulus or rest. This slight variability (i.e., sometimes closer to 10 min, sometimes 15 min) reflects realistic conditions in clinical and research settings, where minor procedural or participant-related delays commonly occur.
By providing both a clear hypothesis and a rationale for the time intervals used, this study addresses a gap in the literature and offers a reproducible methodology for reliably assessing Achilles tendon stiffness in healthy and potentially clinical populations.”
Comments 2:
Provide a detailed rationale for the sample size and explain its implications for reliability studies.
Response 2:
We used Zou’s formula (ICC.Sample.Size package, R) to aim for an ICC of 0.9 with the lower bound of its 95% confidence interval exceeding 0.5, a 5% significance level, and 80% power. Under these parameters, n = 8 participants were required. Each participant contributed multiple repeated measures, enhancing the precision of the ICC estimates. While larger samples may improve generalizability, this repeated-measures design optimizes feasibility and accuracy. We acknowledge that our findings may not fully extend to other populations, and we recommend future research with larger or more diverse samples to confirm and expand upon these results.
Line 212 “These criteria yielded a relatively narrow age range (median: 27.5 years), which may limit generalizability to broader. Furthermore, although the MyotonPRO may be used in tendinopathy cases, our sample included only healthy participants.”
Comments 3:
Include more robust statistical analysis to account for variability and potential outliers.
Response 3:
We appreciate this recommendation. In the revised manuscript, we have taken steps to better account for variability and potential outliers by:
- Adding the interquartile range (IQR) values for the CV, SEM, and MDC metrics (table 3), which provide additional information on the variability and distribution of these measurements.
- Screening for outliers: We initially screened all data using the criterion of values exceeding the mean ± 3 standard deviations; no values met the exclusion criteria. This confirms that our dataset did not contain extreme outliers that could unduly influence our reliability estimates.
- Adding Bland-Altman anylysis
Line 377 “We initially screened all data for extreme outliers (values exceeding the mean ± 3SD), but none met our predefined exclusion criteria. To assess potential systematic bias between repeated measurements for immediate reliability, we compared exactly two measurements at a time (i.e., measurements 1 and 2, 2 and 3, and 1 and 3) using a Bland–Altman analysis (Supplementary Material). For each pair, the mean difference (bias) and 95% limits of agreement were calculated, and a paired t-test was performed to determine whether the bias was statistically significant. The magnitude of the observed bias was then compared to the minimal detectable change (MDC) to evaluate its clinical relevance.”
Line 434 “Formal analysis of potential systematic bias was performed. The Bland-Altman analysis revealed statistically significant negative systematic biases ranging from -5.43 to -11.52 N/m (all p < 0.001) (Supplementary material). However, in all cases, the bias remained within the range of minimal detectable change (MDC = ~15 – ~62 N/m), indicating that the observed differences are unlikely to be clinically meaningful.”
Comments 4:
Expand the discussion of the effects of ADL and SP on Achilles tendon stiffness, linking it to broader clinical applications.
Response 4:
We have expanded the section discussing how activities of daily living (ADL) and sport protocol (SP) might affect tendon stiffness and indicated how this knowledge could inform rehabilitation scheduling or monitoring strategies in clinical contexts.
Line 506 “Additionally, our results revealed that both activities of daily living (ADL) and the simulation of sport had no notable immediate impact on Achilles tendon stiffness. While this suggests a certain resilience of tendon stiffness to single bouts of loading, it also provides practical insight for planning measurement schedules in clinical or research settings. Nonetheless, the cumulative effects of repetitive or prolonged loading over days or weeks remain an important area for further investigation.”
Comments 5:
Revise the limitations section to acknowledge potential biases introduced by using healthy participants
Response 5:
We have added a note explicitly stating that healthy participants may differ from clinical populations in ways that could introduce bias or limit generalizability.
Line 599 „Nevertheless, these findings primarily apply to healthy young adults under controlled conditions, and must be interpreted with caution in clinical scenarios involving older or symptomatic populations.“
Comments 6:
Improve visual data representation, adding plots or graphs to complement the tables.
Response 6:
We appreciate this suggestion and have incorporated a graph (Figure 3) to visually represent the reliability analysis across different testing conditions. The graph provides a clear and immediate comparison of ICC values for each measurement scenario, enhancing the interpretability of the results. By integrating this graphical representation, we aim to complement the tabular data, allowing readers to quickly grasp the variations in reliability across different time frames and conditions.
Comments 7:
Consider performing subgroup analyses or sensitivity analyses to strengthen the findings.
Response 7:
Thank you for this valuable suggestion. In response, we have now included a subgroup analysis comparing the two measurement sites on the Achilles tendon:
- Subgroup Analysis Results: The proximal measurement point yielded an ICC of 0.966 (95% CI: 0.958–0.973), and the distal measurement point yielded an ICC of 0.956 (95% CI: 0.944–0.966) for immediate standardized measurements. These comparable, high ICC values confirm that both regions can be reliably measured using the standardized seated position.
- Although we did not perform formal sensitivity analyses, the robustness of our findings is supported by the subgroup results and the additional reporting of IQR (Table 3.) values for our variability metrics. We acknowledge that sensitivity analyses could further elucidate potential variability under different analytical conditions and will consider this approach in future studies.
Line 428 “3.1 Subgoup and systemic bias analysis
In addition to the overall reliability analyses, we performed subgroup analyses comparing the two measurement sites on the Achilles tendon. The proximal measurement point yielded an intraclass correlation coefficient (ICC) of 0.966 (0.958 – 0.973), while the distal measurement point yielded an ICC of 0.956 (0.944 – 0.966) for immediate SM. These high and comparable ICC values at both sites further support the robustness and consistency of our standardized measurement protocol. Formal analysis of potential systematic bias was performed. The Bland-Altman analysis revealed statistically significant negative systematic biases ranging from -5.43 to -11.52 N/m (all p < 0.001) (Supplementary material). However, in all cases, the bias remained within the range of minimal detectable change (MDC = ~15 – ~62 N/m), indicating that the observed differences are unlikely to be clinically meaningful.”
Thank You

Reviewer 4 Report
Comments and Suggestions for Authors
Review comment
This manuscript entitled “Assessing The Test-Retest Reliability of MyotonPRO for Meas-2 uring Achilles Tendon Stiffness” evaluates the test-retest reliability of the MyotonPRO for measuring Achilles tendon stiffness, focusing on various time intervals, physical activity stimuli, and inter-rater reliability. The manuscript addresses an important topic in sports biomechanics and clinical research, offering a standardized methodology for assessing tendon stiffness. The study’s strengths include detailed experimental procedures, robust statistical analysis, and its practical implications for clinical applications. However, in my opinion, the manuscript has several areas requiring significant revision to improve its clarity, depth, and scientific rigor. These include insufficient justification for sample size, limited generalizability due to the participant cohort, and inadequate exploration of broader implications of findings. Given these concerns, I recommend major revision before the manuscript is suitable for publication.
Specific comments
1. In the abstract part, “Tendon stiffness was measured at two different points (proximal and distal) and at various time frames…”Why were only two points selected for measurement? Could this compromise the generalizability of the findings across the entire Achilles tendon. The reviewer suggest that the authors should consider elaborating on this choice in the abstract. (Line 15)
2. “This study provides a solid foundation for clinical research…”This statement is broad. Can you specify how this foundation contributes to existing methods or clinical interventions. (Line 28)
3. In the introduction part, “Elasticity enables the tendon to store energy in a spring-like manner, while stiffness reduces the extent of elongation…”The distinction between elasticity and stiffness is crucial but could benefit from more precise definitions. Consider citing additional references to provide a clearer biomechanical context. (Line 47–49)
4. “Tendons can elongate by up to 4% of their length without sustaining any damage.” Please use a reference to support this sentence. (Line 49–50)
5. “Shear wave ultrasound elastography (SWUE) has been commonly employed.” Can you provide a direct comparison of SWUE and MyotonPRO in terms of accuracy, reliability, and ease of use? (Line 63)
6. “The objective of this study was to assess the test-retest reliability of the MyotonPRO device…” in my opinion, while the objective is clear, it could be strengthened by explicitly stating how this research addresses gaps in existing knowledge. (Line 93-95)
7. In the Materials and Methods part, “Eight healthy young adults were recruited…” In my opinion, the small sample size and limited diversity (healthy adults under 30) raise concerns about the generalizability of findings. Can you justify why this cohort was chosen and discuss its limitations? (Line 124)
8. “Participants underwent a familiarization session…”Familiarization is a good practice, but it is unclear whether this was sufficient to eliminate learning effects. Were there any metrics authors can confirm this? (Line 181-183)
9. In the discussion part, “The standardized seating position allowed for highly reliable readings…”While this finding is significant, the discussion could benefit from elaborating on how this position compares to other methods, such as prone or standing….
10. “Because this is the first report on measurement errors assessed after different exercise stimuli…”The novelty of the study is appreciated, but can you explore how future studies might build on this work? (Line 284)
11. “The stiffness measurement was only taken at two points…”in my opinion, this limitation is significant and undermines the comprehensiveness of the study. Can you propose solutions or future studies to address this gap? (Line 306–308)
12. In the conclusion part, “providing accurate and consistent measurements of Achilles tendon stiffness.” The conclusion feels repetitive. The reviewer suggests that the author should consider rephrasing to highlight key takeaways and directions for future work. (Line 324-325)
Author Response
Response to Reviewer 4 Comments
|
JFMK-3411868
Assessing The Test-Retest Reliability of MyotonPRO for Measuring Achilles Tendon Stiffness Journal of Functional Morphology and Kinesiology
Reviewer #4
This manuscript entitled “Assessing The Test-Retest Reliability of MyotonPRO for Meas-2 uring Achilles Tendon Stiffness” evaluates the test-retest reliability of the MyotonPRO for measuring Achilles tendon stiffness, focusing on various time intervals, physical activity stimuli, and inter-rater reliability. The manuscript addresses an important topic in sports biomechanics and clinical research, offering a standardized methodology for assessing tendon stiffness. The study’s strengths include detailed experimental procedures, robust statistical analysis, and its practical implications for clinical applications. However, in my opinion, the manuscript has several areas requiring significant revision to improve its clarity, depth, and scientific rigor. These include insufficient justification for sample size, limited generalizability due to the participant cohort, and inadequate exploration of broader implications of findings. Given these concerns, I recommend major revision before the manuscript is suitable for publication.
Comments 1:
In the abstract part, “Tendon stiffness was measured at two different points (proximal and distal) and at various time frames…”Why were only two points selected for measurement? Could this compromise the generalizability of the findings across the entire Achilles tendon. The reviewer suggest that the authors should consider elaborating on this choice in the abstract. (Line 15)
Response 1:
We appreciate this feedback. In the Methods portion of the abstract, we have now briefly justified why we chose to measure only two points (proximal and distal). These points represent commonly used sites for assessing Achilles tendon stiffness, and we aimed to capture the variability near the calcaneal insertion (insertional region) and further up the tendon (mid portion). While we acknowledge that including more measurement sites may provide more granular data, selecting two standardized points helps maintain practicality and consistency for both clinical and research settings.
Line 12 “This study evaluates the test-retest reliability and inter-rater reliability of the MyotonPRO for measuring Achilles tendon stiffness at two strandardized sites over various time frames and settings.”
Comments 2:
“This study provides a solid foundation for clinical research…”This statement is broad. Can you specify how this foundation contributes to existing methods or clinical interventions. (Line 28)
Response 2:
We have revised the concluding sentence in the abstract to clarify the specific contributions of our findings to clinical practices and research—namely, how our standardized protocol and reliability data can help in monitoring tendon adaptations, guiding rehabilitation programs, and informing clinical decision-making related to Achilles tendon care.
Line 26 “When used in a standardized position, the MyotonPRO demonstrates reliable repeated measurements of Achilles tendon stiffness. This protocol provides a foundation for clinical research and rehabilitation by clarifying expected reliability across minutes, days, and weeks, thus aiding clinicians and researchers in monitoring tendon adaptations and making evidence-based decisions.”
Comments 3:
In the introduction part, “Elasticity enables the tendon to store energy in a spring-like manner, while stiffness reduces the extent of elongation…”The distinction between elasticity and stiffness is crucial but could benefit from more precise definitions. Consider citing additional references to provide a clearer biomechanical context. (Line 47–49)
Response 3:
We have expanded the definitions of elasticity and stiffness to emphasize their biomechanical differences and have added references [7] for clarity.
Line 113 „In general, tendons can elongate by up to 4% of their length without sustaining any damage. However, if resistance to elongation is inadequate, elongation between 4% and 8% may lead to the breakdown of collagen cross-links. [7] This can lead to structural changes that contribute to the development of tendinopathy. When elongation exceeds a critical threshold (above 8%), collagen fibers may undergo macroscopic failure, potentially leading to Achilles tendon rupture and complete loss of function.“
- H.-C. Wang, “Mechanobiology of tendon,” J Biomech, vol. 39, no. 9, pp. 1563–1582, 2006. DOI: 10.1016/j.jbiomech.2005.05.011
Comments 4:
“Tendons can elongate by up to 4% of their length without sustaining any damage.” Please use a reference to support this sentence. (Line 49–50)
Response 4:
We have added a citation [7] to support the statement about the 4% elongation threshold.
Line 113 „In general, tendons can elongate by up to 4% of their length without sustaining any damage. However, if resistance to elongation is inadequate, elongation between 4% and 8% may lead to the breakdown of collagen cross-links. [7] This can lead to structural changes that contribute to the development of tendinopathy. When elongation exceeds a critical threshold (above 8%), collagen fibers may undergo macroscopic failure, potentially leading to Achilles tendon rupture and complete loss of function.“
- H.-C. Wang, “Mechanobiology of tendon,” J Biomech, vol. 39, no. 9, pp. 1563–1582, 2006. DOI: 10.1016/j.jbiomech.2005.05.011
Comments 5:
“Shear wave ultrasound elastography (SWUE) has been commonly employed.” Can you provide a direct comparison of SWUE and MyotonPRO in terms of accuracy, reliability, and ease of use? (Line 63)
Response 5:
We have included a brief comparison in the paragraph discussing SWUE and MyotonPRO, highlighting specific differences in operator skill requirement, accuracy, and device portability.
139 „Compared to more operator-dependent modalities (e.g., ultrasound elastography), the MyotonPRO requires less specialized training and reduces user-dependent variability. Its portability and straightforward setup make it a convenient tool for researchers and clinicians to measure soft-tissue stiffness quickly and accurately.“
Comments 6:
“The objective of this study was to assess the test-retest reliability of the MyotonPRO device…” in my opinion, while the objective is clear, it could be strengthened by explicitly stating how this research addresses gaps in existing knowledge. (Line 93-95)
Response 6:
We have strengthened the statement of purpose to explicitly mention how our research addresses previously unexamined time intervals and activity stimuli, thereby filling a gap in the literature.
Line 174 “Considering the lack of established reliability data, the objective of this study was to (1) assess the test-retest reliability of the MyotonPRO in measuring Achilles tendon stiffness using a newly established standardized position. (2) We also aimed to determine whether different time intervals (ranging from seconds to weeks) influence measurement reliability, and (3) we evaluated how physical activity stimuli (activities of daily living and sport-like exercises) affect short-term reliability. Furthermore, (4) we explored the inter-rater reliability of the MyotonPRO in this context.”
Comments 7:
In the Materials and Methods part, “Eight healthy young adults were recruited…” In my opinion, the small sample size and limited diversity (healthy adults under 30) raise concerns about the generalizability of findings. Can you justify why this cohort was chosen and discuss its limitations? (Line 124)
Response 7:
We selected eight healthy adults under 30 based on Zou’s formula for ICC estimation, which indicated n = 8 would be sufficient for our repeated-measures design. By using healthy, younger participants, we aimed to minimize variability unrelated to our primary outcome (i.e., Achilles tendon stiffness) and to focus on establishing the test-retest reliability of the MyotonPRO under controlled conditions. We acknowledge that this sample may not reflect broader clinical populations—particularly older adults or those with tendinopathy—and thus limits generalizability. Future research with larger, more diverse cohorts, including older individuals and symptomatic participants, is encouraged to expand upon and validate these findings.
Line 212 These criteria yielded a relatively narrow age range (median: 27.5 years), which may limit generalizability to broader. Furthermore, although the MyotonPRO may be used in tendinopathy cases, our sample included only healthy participants.
Comments 8:
“Participants underwent a familiarization session…”Familiarization is a good practice, but it is unclear whether this was sufficient to eliminate learning effects. Were there any metrics authors can confirm this? (Line 181-183)
Response 8:
Familiarization aimed to ensure that participants understood the protocols and measurement procedures, thus reducing variability from novice error. During pilot sessions, we observed that participants could perform SP protocol and ADL protocol and maintain the standardized position. We did not collect separate quantitative metrics (e.g., slope of improvement) specifically for learning effects.
Comments 9:
In the discussion part, “The standardized seating position allowed for highly reliable readings…”While this finding is significant, the discussion could benefit from elaborating on how this position compares to other methods, such as prone or standing….
Response 9:
We agree that discussing alternative positions is valuable. We have added a statement indicating that future studies could directly compare the seated position with prone or standing protocols to identify potential advantages in terms of participant comfort, reduced muscle tension, or ease of landmark identification. This comparison would clarify whether seated measurement truly yields superior reliability compared to other standard methods.
Line 486 Nevertheless, future studies could compare seated measurements to prone or standing methods to determine whether the seated position confers additional benefits in terms of participant comfort, reduced muscle tension, or ease of landmark identification.
Line 502 “Muscle tension may also influence stiffness measurements, as any residual activation in the triceps surae could slightly alter the measured tendon properties. By seating participants with knees bent at 90° and encouraging full relaxation, we attempted to minimize this confounder. However, more sensitive electromyographic (EMG) monitoring might be employed in future work to ensure negligible muscle activation during testing. Additionally, our results revealed that both activities of daily living (ADL) and the simulation of sport had no notable immediate impact on Achilles tendon stiffness. While this suggests a certain resilience of tendon stiffness to single bouts of loading, it also provides practical insight for planning measurement schedules in clinical or research settings. Nonetheless, the cumulative effects of repetitive or prolonged loading over days or weeks remain an important area for further investigation.”
Comments 10:
“Because this is the first report on measurement errors assessed after different exercise stimuli…”The novelty of the study is appreciated, but can you explore how future studies might build on this work? (Line 284)
Response 10:
We have expanded the discussion to suggest that subsequent research could investigate additional types of loading (e.g., plyometric or sprint-based protocols) or longer-term, repetitive stresses. This would help determine whether the current findings generalize to more dynamic or sustained exercise, thereby enhancing our understanding of tendon stiffness adaptations.
Line 486 Nevertheless, future studies could compare seated measurements to prone or standing methods to determine whether the seated position confers additional benefits in terms of participant comfort, reduced muscle tension, or ease of landmark identification.
Line 577 Future work might explore other sampling sites, employ alternative exercise stimuli, and include symptomatic individuals to better elucidate the MyotonPRO’s full potential in rehabilitation and performance contexts.
Comments 11:
“The stiffness measurement was only taken at two points…”in my opinion, this limitation is significant and undermines the comprehensiveness of the study. Can you propose solutions or future studies to address this gap? (Line 306–308)
Response 11:
We acknowledge that sampling only two measurement sites limits the ability to capture the full stiffness profile of the Achilles tendon. However, we selected these two point specifically because they reflect the key anatomical regions most commonly implicated in Achilles tendinopathies (i.e., insertional and mid–portion area). We recommend that future research either include more measurement points (e.g., every 1–2 cm along the tendon) or combine MyotonPRO with imaging techniques (e.g., ultrasound) for a more comprehensive stiffness map. This multi-site approach would provide deeper insights into regional variations in tendon stiffness.
Line 428 3.1 Subgoup and systemic bias analysis
In addition to the overall reliability analyses, we performed subgroup analyses comparing the two measurement sites on the Achilles tendon. The proximal measurement point yielded an intraclass correlation coefficient (ICC) of 0.966 (0.958 – 0.973), while the distal measurement point yielded an ICC of 0.956 (0.944 – 0.966) for immediate SM. These high and comparable ICC values at both sites further support the robustness and consistency of our standardized measurement protocol. Formal analysis of potential systematic bias was performed. The Bland-Altman analysis revealed statistically significant negative systematic biases ranging from -5.43 to -11.52 N/m (all p < 0.001) (Supplementary material). However, in all cases, the bias remained within the range of minimal detectable change (MDC = ~15 – ~62 N/m), indicating that the observed differences are unlikely to be clinically meaningful.
Comments 12:
In the conclusion part, “providing accurate and consistent measurements of Achilles tendon stiffness.” The conclusion feels repetitive. The reviewer suggests that the author should consider rephrasing to highlight key takeaways and directions for future work. (Line 324-325)
Response 12:
We have revised the conclusion to avoid repetition and to emphasize the primary takeaways. We also added specific suggestions for future research (e.g., testing symptomatic individuals, investigating diverse activity stimuli, and expanding measurement sites), thereby offering clearer directions for clinicians and researchers seeking to apply or build upon our findings.
Line 571 “Within these parameters, the MyotonPRO can be confidently employed to capture tendon stiffness values in a consistent manner. However, additional research is warranted to determine the device’s performance in diverse patient groups and real-world clinical settings.
In conclusion, while these findings underscore the reliability of MyotonPRO measurements for Achilles tendon stiffness in healthy young adults, they should be interpreted with caution when applying them to clinical populations. Future work might explore other sampling sites, employ alternative exercise stimuli, and include symptomatic individuals to better elucidate the MyotonPRO’s full potential in rehabilitation and performance contexts.”

Round 2
Reviewer 2 Report
Comments and Suggestions for Authors
In response to my review comments, the authors have implemented comprehensive revisions and provided valuable explanations and clarifications. These additions are both reasonable and effective, significantly enhancing the overall quality of the manuscript.
Reviewer 3 Report
Comments and Suggestions for Authors
Dear Authors,
I would like to thank you for taking the time and effort to implement all the suggested revisions with such diligence and integrity. In my opinion, your paper is now even more complete. I have no further comments to add.
Reviewer 4 Report
Comments and Suggestions for Authors
The reviewers thank the authors for their efforts in the revision. I think this manuscript has reached the acceptance standard.
Comments on the Quality of English LanguageNone.